# Neoadjuvant Chemotherapy before Nephroureterectomy in High-Risk Upper Tract Urothelial Cancer: A Systematic Review and Meta-Analysis

**DOI:** 10.3390/cancers14194841

**Published:** 2022-10-04

**Authors:** David Oswald, Maximilian Pallauf, Susanne Deininger, Peter Törzsök, Manuela Sieberer, Christian Eiben

**Affiliations:** 1Department of Urology and Andrology, University Hospital Salzburg, Paracelsus Medical University, 5020 Salzburg, Austria; 2Department of Urology, The James Buchanan Brady Urological Institute, The Johns Hopkins University School of Medicine, Baltimore, MD 21205, USA

**Keywords:** upper urinary tract urothelial carcinoma, upper urinary tract urothelial cancer, upper tract urothelial carcinoma, upper tract urothelial cancer, UTUC, neoadjuvant chemotherapy, preoperative chemotherapy, neoadjuvant Immunotherapy, preoperative immunotherapy, neoadjuvant check point inhibitor, preoperative check point inhibitor

## Abstract

**Simple Summary:**

In muscle-invasive bladder cancer, platinum-based neoadjuvant chemotherapy is a well-established concept, since it contributes to pathological downstaging and improves overall survival. Immunotherapy has become increasingly important in adjuvant and palliative treatment for metastatic urothelial carcinoma, and phase II studies have shown assuring data in neoadjuvant treatment before cystectomy. Although upper urinary tract urothelial cancer usually presents as more invasive at diagnosis than bladder cancer, there is no recommendation for neoadjuvant therapy before nephroureterectomy. This meta-analysis comprises eleven comparative trials on neoadjuvant chemotherapy in this setting and analyzes pathological downstaging as well as oncological outcomes. However, no comparative studies investigating immunotherapy in upper tract urothelial cancer were found. The pathological downstaging and complete response were significantly improved in patients who received neoadjuvant chemotherapy. The data also suggested a benefit in overall survival and progression-free survival in these patients. Prospective studies are needed to confirm these findings and assess the role of immunotherapy in this setting.

**Abstract:**

Neoadjuvant chemotherapy is a well-established concept in muscle-invasive bladder cancer with known advantages in overall survival. Phase II trials show encouraging response rates for neoadjuvant immunotherapy before radical surgery in urothelial cancer. There is no recommendation for neoadjuvant therapy in upper tract urothelial carcinoma before nephroureterectomy. Our aim was to assess the available data on neoadjuvant chemotherapy and immunotherapy before nephroureterectomy in patients with high-risk upper tract urothelial carcinoma in terms of pathological downstaging and oncological outcomes. Two investigators screened PubMed/Medline for comparative trials in the English language. We identified 368 studies and included eleven investigations in a systematic review and meta-analysis for neoadjuvant chemotherapy and control groups. There were no comparative trials investigating immunotherapy in this setting. All 11 studies reported on overall pathological downstaging with a significant effect in favor of neoadjuvant chemotherapy (OR 5.17; 95%CI 3.82; 7.00). Pathological complete response and non-muscle invasive disease were significantly higher in patients receiving neoadjuvant chemotherapy (OR 12.07; 95%CI 4.16; 35.03 and OR 1.62; 95%CI 1.05; 2.49). Overall survival and progression-free survival data analysis showed a slight benefit for neoadjuvant chemotherapy. Our results show that neoadjuvant chemotherapy is effective in downstaging in upper urinary tract urothelial carcinoma. The selection of patients and chemotherapy regimens are unclear.

## 1. Introduction

Urothelial carcinoma (UC) is the 6th most common tumor entity in developed countries. Upper urinary tract urothelial carcinoma (UTUC) accounts for 5–10% of all UCs. The rest mostly comprises bladder cancer (BC), accounting for approximately 90–95%, as well as a few cases of urethral cancer [1,2]. At first diagnosis, approximately 60% of UTUC shows invasive pathology [3,4], as opposed to only 25% in BC [5]. Prognosis is poor, especially for high grade and advanced diseases with overall 5-year disease-specific survival (DSS) rates ranging from 57% to 73% [3,6,7,8].

There are different environmental and pathological risk factors influencing oncological outcomes in UTUC [3,8,9,10,11,12,13,14,15]. High-risk disease has been defined by the European Association of Urology as any criteria of the following: multifocal disease, tumor size ≥2 cm, high-grade cytology, high-grade ureterorenoscopic biopsy, local invasion on computed tomography (CT), hydronephrosis, previous radical cystectomy for high grade BC, and variant histology [2].

Diagnostics rely on radiologic imaging, cytology and ureterorenoscopy (URS) with biopsy [2,10]. In case of localized, high-risk UTUC, radical nephroureterectomy (RNU) with template lymph node dissection is recommended as the therapy of choice, whilst low-risk tumors can be approached with kidney-sparing strategies [2].

In patients with muscle-invasive diseases, adjuvant platinum-based chemotherapy is currently recommended, since it increases disease-free survival [2,16]. However, there is no actual recommendation on neoadjuvant therapy for UTUC in the European Association of Urology (EAU) Guidelines, aside from a flow chart implying +/− “perioperative” platinum-based chemotherapy for high-risk UTUC [2]. In localized, clinically lymph node negative muscle-invasive bladder cancer (MIBC), neoadjuvant cisplatin-based chemotherapy is a well-known and guideline-recommended strategy, because it improves overall survival (OS) by 5–8% at 5 years postoperatively [17,18,19]. Phase II trials using checkpoint-inhibitors (CPIs) pembrolizumab and atezolizumab as neoadjuvant approach before radical surgery also have shown favorable results in terms of response and pathological downstaging with an acceptable safety profile [20,21].

Although management of UTUC often is derived from findings in BC [22,23,24], it is well known that UTUC acts differently than lower tract tumors and shows distinct molecular patterns [25,26]. Biologically, a predominance of the luminal subtype of UC as well as a high FGFR-3 density have been shown [27,28,29,30], which might indicate worse responses to checkpoint inhibition [31]. Adversely, Moss et al. found an upregulation of CPI target CTLA4, PD1 and PDL1 expression in a subset of patients with clinically aggressive diseases, which might indicate an increased effect of immunotherapy in this collective [29]. However, this finding has to be interpreted with great care since only 31 patients were included in the overall analysis, with even less patients included in the sub-groups. Balar et al. investigated the CPI atezolizumab in a first line setting for cisplatin-ineligible locally advanced and metastatic UC. They found higher response rates in patients with UTUC as a primary than patients with lower tract primaries [32].

On the other hand D’Andrea et al. showed comparable results in terms of pathological downstaging for neoadjuvant chemotherapy in UTUC and MIBC in a multi-centric retrospective analysis [33].

The rationale for neoadjuvant therapy in UTUC comprises pathological downstaging, treatment of micro-metastases and eligibility for cisplatin and higher possible doses of chemotherapy due to better kidney function before RNU, respectively [34]. Downstaging to T0 seems to be the best surrogate for survival [35], as for MIBC [36].

An obvious benefit of neoadjuvant therapy regimens compared to the adjuvant setting is eligibility for chemotherapy, which decreases distinctly after RNU due to renal insufficiency [37,38,39,40]. In terms of drawbacks, there might be a delay to definitive surgery depending on local surgery waiting periods, which might have an impact especially on non-responders [41,42]. Since there usually is no pathological evidence of muscular invasion before surgery [43], patients with T1 disease or lower might be overtreated with neoadjuvant therapy [19].

Data for neoadjuvant therapy in UTUC has always been scarce. Nevertheless, due to increasing insights in chemotherapy and experiences with new agents in BC, preoperative systemic therapy has been suggested to be an option for some patients [8,33,44,45]. In 1995 Igawa et al. showed a CR (complete response) rate of 13% and an overall response rate of 53% in a collective of 15 patients undergoing neoadjuvant chemotherapy (NAC) with methotrexate, vinblastine, adriamycin/epirubicin and cisplatin (MVAC/MVEC) [46] before RNU. A phase II trial using sequential platinum-containing NAC for UC included five cases of UTUC. The results showed good tolerability and pathological downstaging in 60% [35]. In daily practice, the proportion of patients undergoing RNU treated with NAC is low, but increasing [8,47].

Our aim was to assess the available data on different neoadjuvant concepts, including chemotherapy and CPIs, in a systematic review and meta-analysis.

## 2. Material and Methods

The systematic review and meta-analysis was performed according to the 2015 PRISMA-P Consensus statement [48]. The study was registered in the International prospective register of systematic reviews (PROSPERO) [49] with the unique identification number CRD42022327241.

### 2.1. Eligibility Criteria

Inclusion criteria: We included controlled observational trials, controlled retrospective trials and randomized controlled trials in English language. In terms of PICO criteria (participants, interventions, comparators, outcomes) studies must research neoadjuvant therapy regimens including either chemotherapy or immunotherapy in patients with high-risk UTUC undergoing RNU. Clinically positive locoregional lymph nodes were allowed. Predominant urothelial tumor histology throughout the study population was required for inclusion. Oncological outcomes in terms of pathological or clinical response (CR, partial remission (PR), stable disease (SD), progressive disease (PD)); pathological downstaging and/or progression free survival (PFS), DSS and OS, respectively, should have been reported. The data cutoff was 02/2022.

Exclusion Criteria: Articles in languages other than English, editorials, letters to the editor, and case reports were excluded from the study. Additionally, single-arm studies without a comparator were excluded from systematic review. Collectives including patients with non-locoregional metastases were excluded from analysis.

### 2.2. Information Sources

PubMed/Medline was searched systematically. Reference lists of screened publications were also screened for additional publications, which could not be identified by the pre-specified keywords.

### 2.3. Search Strategy

The pre-specified keywords were used to search the pre-specified databases in different combinations, including a synonym for UTUC and neoadjuvant therapy each.

### 2.4. Study Records

Data management: All identified studies were recorded using Mendeley [50]. The data retrieved during data extraction phase was saved using Microsoft Excel [51] spread sheets.

Selection Process: Two independent reviewers screened the databases according to the keywords for according studies, excluded duplicates and checked for eligibility by review of title and abstract. Full text read was performed for unclear cases. The selection of both authors was then compared for discrepancies. In such cases, a third reviewer was asked to aid in decision-making. A detailed flowchart of the selection process is depicted in Figure 1.

Data Collection Process: Data was extracted according to a pre-specified Excel sheet by both authors to avoid typing errors. Recording of additional interesting data, which had not been pre-specified, was allowed. In such a case, the spreadsheet was adapted and the items in question had to be checked, with all studies already screened for missed data.

### 2.5. Data Items

Baseline parameters: Pre-specified baseline data included type of neoadjuvant therapy, dose, cycles, number of patients, tumor location, multifocality, gender, tumor size, smoking status, hydronephrosis, baseline glomerular filtration rate (GFR), Eastern Cooperative Oncology Group (ECOG) performance score, previous or concurrent BC, diagnostic measures, clinical/radiological (re-)staging according to TNM classification [52], staging modalities, time to surgery from diagnosis and from end of chemotherapy, performed lymphadenectomy and open or laparoscopic surgery.

Outcomes and prioritization: Clinical and pathological response (CR, PR, SD, PD), pathological staging according to TNM classification [52], pathological downstaging, lymphovascular invasion (LVI), PFS, DSS, OS, and adverse events (per category and per Clavien-Dindo classification) were recorded to assess them effectivity and safety. The main prioritization was to analyze effectivity of neoadjuvant therapy in terms of pathological and clinical response as well as long time oncological outcome.

Due to extensive differences in data presentation throughout the publications, some outcomes were grouped for analysis. For example, some studies only presented data on T3 and T4 tumors as pooled results, so we grouped data into a T3/T4 category for the rest of the included studies as well, in order to achieve an overall more representative comparison.

OS, PFS and DSS data were only presented in a proportion (percent), but not in total numbers throughout all studies. For these parameters, rounding errors have to be accounted for.

### 2.6. Risk of Bias Assessment

Risk of bias assessment was performed according to the ROBINS-I tool for non-randomized studies [53] after study selection. It showed overall low risk of bias for the select papers.

### 2.7. Data Synthesis

A meta-analysis based on 11 studies was created to compare the effects of “study” vs. “control”. Studies with double zeros are included in the meta-analysis using a continuity correction. The effect size was estimated based on odds-ratios (ORs) and hazard ratios (HRs) including their corresponding two-sided 95% confidence intervals (CIs). Based on the estimated heterogeneity between the studies, which was assessed using Higgins I^2^ and Cochran’s Q, a common fixed effect or random effects model was calculated. Funnel plots and Egger’s test for asymmetry were used to assess potential publication bias.

A *p*-value < 0.05 was taken as the uncorrected statistical significance level (two-sided), therefore, all inferential results are only descriptive. For statistical analysis, the statistical computing software R Version 4.1.2 (R Foundation for Statistical Computing, Vienna, Austria; URL: http://www.R-project.org, accessed on 22 April 2022) was used. For conducting the meta-analysis, the R packages meta [54] and metafor were used.

Data not appropriate for quantitative synthesis were presented narratively, in the context of current literature.

## 3. Results

### 3.1. Search Results

Overall, 368 studies were found using the pre-specified keywords. No additional publications were identified through other sources. After elimination of duplicates, 297 studies remained for title and abstract screening. A full-text read was performed on 42 studies. Finally, eleven studies met the eligibility criteria and were included for systematic review and meta-analysis.

### 3.2. Baseline Parameters

Baseline study characteristics are shown in Table 1. The analyzed studies comprised a broad range of different chemotherapy agents, as well as schedules. No controlled studies were available for immunotherapy in the neoadjuvant setting for high-risk UTUC. Some publications excluded clinically locoregional lymph node positive disease from analysis (*n* = 5), while others included clinically locoregional lymph node positive disease (*n* = 3) or only analyzed tumors with clinically locoregional lymph node invasion (*n* = 3).

The selected baseline population characteristics, as far as reported, are depicted in Table 2. Clinical tumor and lymph node staging were similar throughout study and control groups. There was a tendency towards laparoscopic surgery in the NAC groups.

### 3.3. Pathological Downstaging

Pathological downstaging was assessed in different ways depending on the available data. Firstly, meta-analyses of pathological T stages and N stages in patients with and without neoadjuvant chemotherapy are presented as a surrogate marker of difference in downstaging. Secondly, the clinical to pathological downstaging, if available, was analyzed. Thirdly, the overall pathological downstaging and LVI as presented by the studies were assessed.

Six studies reported on CR in terms of pathological T0 stage. Meta-analysis showed a significantly higher proportion of pathological CR in the populations that received neoadjuvant therapy (OR 12.07; 95% CI 4.16; 35.03) (Figure 2).

Five studies presented a pathological T1 stage of neoadjuvant chemotherapy and control groups (non-muscle invasive). Results showed borderline significantly higher numbers within the preoperatively treated patients (OR 1.62; 95% CI 1.05; 2.49) (Figure 3).

Five studies reported on a pathological T2 stage (muscle-invasive). No significant difference could be shown in between the groups (OR 1.15; 95% CI 0.73; 1.80) (Figure 4).

Six studies presented data on clinical T3 and T4 stages, which were grouped as locally invasive. There were significantly higher numbers of advanced disease stages within the control groups, which had not received neoadjuvant therapy (OR 0.27; 95% CI 0.19; 0.38) (Figure 5).

Five studies assessed pathologically positive lymph nodes (pN+). There was no significant difference between the study and the control groups (OR 0.90; 95% CI 0.56; 1.44) (Figure 6).

Only two studies explicitly reported on reduction of clinical to pathological stage throughout both groups. These showed a significant difference in favor of neoadjuvant therapy (OR 11.98; 95% CI 2.04; 70.33) (Figure 7).

The same studies also presented data on the downstaging of clinically positive to pathologically negative lymph node stage (OR 8.49; 95% CI 2.76; 26.14) (Figure 8).

All studies assessed overall pathological downstaging with a significant difference in favor of neoadjuvant therapy (OR 5.17; 95% CI 3.82; 7.00) (Figure 9).

Seven studies reported on lymphovascular invasion (LVI) in a pathological specimen. A significant difference, with a higher proportion of LVI positive tumors within the control groups, could be shown (OR 0.48; 95% CI 0.38; 0.62) (Figure 10).

### 3.4. Oncological Outcomes

Two studies presented three-year OS, showing a significant benefit for the study groups (OR 3.80; 95% CI 1.80; 8.01) (Figure 11).

Six studies reported on five-year OS, also showing a significant difference in favor of the study groups (OR 2.46; 95% CI 1.45; 4.16) (Figure 12).

In terms of five-year DSS, five studies presented information showing a significant benefit in favor of neoadjuvant therapy (OR 2.53; 95% CI 1.78; 3.59). However, Egger’s graph was significant for publication bias (*p* = 0.031) (Figure 13).

Three studies reported on five-year PFS with a positive trend in favor of neoadjuvant therapy, but without reaching significance (OR 1.78; 95% CI 0.92; 3.44) (Figure 14).

Two studies reported PFS data adjusted for multivariate analysis (Intuition for Inverse Probability of Treatment Weighting (IPTW) adjusted PFS). Meta-analysis showed a significant risk reduction in disease progression in patients treated with NAC (HR 0.42; 95% CI 0.19; 0.91) (Figure 15).

Four studies reported DSS data adjusted for multivariate analysis (IPTW adjusted DSS). A significant risk reduction with NAC could be shown in the analysis (HR 0.44; 95% CI 0.32; 0.61) (Figure 16).

Six studies reported OS data adjusted for multivariate analysis (IPTW adjusted OS). A significant risk reduction with NAC could be shown in the analysis (HR 0.51; 95% CI 0.40; 0.66). However, Egger’s graph for publication bias was positive (*p* = 0.034) (Figure 17).

## 4. Discussion

We assessed eleven comparative studies, which represents an update to the latest work from Leow et al. 2020, who included seven controlled studies [66]. All studies comprised representative populations in terms of age and gender. UTUC is known to be more frequent in men, with a male to female ratio of 2:1 [67]. The same is true for the location of the primary tumor with a known ratio of 2:1 in the renal pelvis versus the ureter [68]. Clinical staging data was presented descriptively with balanced numbers between the groups, and therefore represents a fair starting point for downstaging analysis. Due to differences in timelines, markedly more laparoscopic and robotic RNUs were performed in patients who received neoadjuvant chemotherapy. We did not present further patient characteristics due to the extensive differences in data reporting and presentation.

Pathological downstaging was assessed differently throughout the studies, but all of them showed a significant difference in favor of neoadjuvant therapy (OR 5.17). However, details of assessment, such as clinical and pathological staging data per patient, were not presented throughout all trials. In terms of the T0 stage, six studies showed significantly higher numbers of pathologically complete responses within the study groups. The same was true for the T1 stage, with significantly higher numbers of non-invasive tumors within the study groups, albeit less unequivocal. Reversely significantly higher numbers of locally advanced tumors (T3/4) could be found in the control groups. No differences could be shown in localized invasive disease (T2). Pathological T0 disease is a clear surrogate for complete response and the pathological T3/4 stage (locally advanced disease) helps to address the response or failure to response. However, T1 (non-muscle-invasive) and T2 (muscle-invasive, non-locally advanced disease) are more difficult to interpret. We believe they might have a role in the assessment of partial response. For example, our finding of more T1 tumors within the study group could be a hint for increased partial response within these patients. However, this notion is based on balanced selection of patients’ initial clinical tumor stage and subjected to major selection bias. It might be interesting to gather this data on an individual patient basis, thereby a thorough analysis of actual downstaging as well as degree of downstaging is possible.

Two studies explicitly presented data on pathological downstaging from clinical staging, with a significant benefit for the study groups. Additionally, LVI was significantly more often present within control groups, which had not received neoadjuvant therapy.

Altogether, these results show a relevant benefit in terms of tumor downstaging after NAC before RNU. This is especially true since clinical tumor stages were distributed rather equally throughout the groups (Table 2).

Interestingly, there was no significant difference in pathologically positive lymph nodes with or without NAC. However, two studies presented data explicitly on downstaging from clinical positive to negative lymph nodes with significant benefit in favor of the study groups. These results highlight the need for prospective trials with more differentiated study populations, which distinguish clinically locoregional lymph node positive and negative populations. This question is crucial considering MIBC, in which clinically lymph node positive disease is excluded from neoadjuvant therapy according to the EAU guideline [17], and inductive chemotherapy is to be considered.

Three- and five-years OS data showed a benefit for neoadjuvant therapy. This was consistent with our results in the IPTW adjusted outcomes. However, Egger’s Test yielded a positive result for publication bias, so these results should be interpreted with caution. A five-years DSS also showed a benefit for neoadjuvant therapy, and Egger’s Test was positive as well, so we would recommend waiting for more solid data on this issue. For five-years PFS, the study results did not show significant results, although there seemed to be a trend towards increased PFS within the study groups. An analysis of IPTW-adjusted PFS showed significant risk reduction for patients who received neoadjuvant therapy.

All in all, retrospective data on pathological downstaging shows a benefit for NAC, whilst oncological follow-up data was too equivocal to allow a final conclusion. Nevertheless, data on OS and PFS seems promising.

We could not identify any controlled trials investigating CPIs in the neoadjuvant setting for UTUC. There are several phase II trials showing significantly increased complete response rates for neoadjuvant therapy with different CPIs in invasive UC before radical surgery, ranging from 31 to 46% [20,21,69,70]. This effect was more pronounced in PD-L1 positive patients in all trials. Notably, in some studies, patients with UTUC were included, but subgroup analyses are missing [21,70].

The CPI Nivolumab showed significantly increased PFS in high-risk muscle-invasive UC patients, including those with UTUC, in an adjuvant setting [71]. However, subgroup analysis still has to be conducted before treatment recommendations can be given, according to EAU guidelines [2].

Some single-arm trials present data on neoadjuvant CPI therapy for UTUC cohorts. The PURE-02 study comprises a collective of only ten patients undergoing neoadjuvant therapy with three cycles of pembrolizumab. Only one patient achieved CR, and two patients progressed and received chemotherapy. The authors concluded that pembrolizumab as single-agent therapy is not suitable in this setting [72].

Additionally, multiple trials are in progress or being set up for neoadjuvant concepts in MIBC [73]. The EV-103 study (ClinicalTrials.gov Identifier: NCT03288545) includes experimental arms investigating the drug-antibody conjugate Enfortumab-Vedotin with or without the CPI Pembrolizumab. In SURE-01, the same is done for a similar drug, Sacituzumab-Govitecan, with or without Pembrolizumab for patients who cannot or do not want to receive cisplatin-based therapy (ClinicalTrials.gov Identifier: NCT05226117). However, neither of these trials includes patients with UTUC. All in all, the role of neoadjuvant immunotherapy in UTUC has yet to be determined.

Considering the high amount of FGFR-3 expression in UTUC, another targeted agent class to be investigated in the future is FGFR-3 inhibitors, such as erdafitinib. It has shown efficacy in receptor positive UC and is approved for therapy in locally advanced and metastatic UC [17,74].

Our results represent an update to previous systematic analyses attending to the same matter [66,75,76,77]. Yang et al. found a significant improvement of DSS with a HR of 0.25 (95% CI 0.06–0.61) in patients receiving NAC for UTUC before RNU [77]. Gregg et al. found an OS benefit with an HR of 0.36 (95% CI 0.19–0.69, *p* = 0.002), albeit only two retrospective comparative trials could be included [76]. Kim et al. investigated four retrospective comparative trials in their analysis. They found an overall OS benefit for NAC with a pooled HR of 0.46 (95% CI 0.27–0.79, *p* = 0.005). Furthermore, they showed improved DSS with a pooled HR of 0,41 (95% CI 0.26–0.65, *p* = 0.0001) and PFS with a pooled HR of 0.53 (95% CI 0.39–0.73, *p* ≤ 0.0001) within the study groups. The pooled odds ratio (OR) for pathological downstaging was 0.21 (95% CI 0.09–0.06, *p* = 0.004), indicating a strong advantage of NAC. In contrast to our analysis, the proportion of pathologically positive lymph nodes was significantly lower within the study groups [75]. Leow et al. performed the biggest analysis so far, comprising seven comparative and nine single-arm studies. Five of the single-arm studies were prospective in nature, the rest had a retrospective design. The pooled HR of six studies for OS was 0.44 (95% CI 0.32–0.59, *p* < 0.001). DSS was improved with a pooled HR of three studies of 0,38 (95% CI 0.24–0.61, *p* < 0.001). The pooled pathological CR rate was 11% (95% CI 8–15%) and the pooled rate of downstaging was 33% (95% CI 14–52%) [66]. In contrast to the work of Leow et al., who also analyzed an adjuvant treatment regimen, we only concentrated on neoadjuvant therapies. Additionally, we looked at controlled studies only. In return, our meta-analysis included multiple different parameters such as lymph node state, lymphovascular invasion and different categories of pathological downstaging according to the different trial designs. Even though prospective evidence is still needed to draw tangible conclusions for clinical decision making, our data help to pose the correct questions in these trials and find the appropriate patient collective, which might profit from neoadjuvant therapy. As an example, our finding of equivalent numbers of patients with pathologically positive lymph nodes in both groups highlights the necessity of stratification for clinically positive lymph nodes. Concerning OS and CSS data, we analyzed three-year and five-year data, depending on the available results. Additionally, we analyzed PFS with two different analyses, suggesting a benefit for NAC. Furthermore, our work highlights the need for additional research on immunotherapy in this setting.

## 5. Limitations

The major limitations of our results are the retrospective nature of the analyzed trials, the inhomogeneous study populations and treatment regimens, as well as major differences in data presentation. Most notably, the criteria for administration of NAC throughout the different trials cannot be reproduced, which might represent a major selection bias.

A challenge in UTUC as compared to BC is that there rarely is representative histology before surgery. Histology acquired by URS and forceps-biopsy usually only helps to determine tumor grade and get a visual impression of tumor architecture [26]. Biopsy was shown to be highly accurate in the determination of grading, but not staging with significant correlation of high-grade diseases on biopsy with high-grade and muscle-invasive features of the pathological specimen [78,79,80]. Clinical staging is mostly radiological and relies on imaging (usually CT) [81,82]. A partial pathological response to neoadjuvant treatment is therefore difficult to assess, since it is impossible to know whether a patient had pT1 diseases in the first place or because of downstaging after neoadjuvant treatment. None of the investigated studies reported on radiological re-staging after neoadjuvant therapy before surgery, which would have helped to assess the radiological response.

CT urography represents the most accurate means of clinical staging with correct prediction of the pathological stage in 88% in a case series [83]. The staging of organ confined diseases was correct in 97%, but only 67% of cases in locally advanced diseases in a different collective [84]. Thus, it might be a worthwhile strategy for future trials to repeat radiographic staging after neoadjuvant therapy for assessment of the response.

Due to the retrospective data analysis, chemotherapy regimens varied widely across the studies, and often different schemata with different number of cycles were used within the same populations *(*Table 2*)*. Nevertheless, platinum-containing regimens were predominating, with Gemcitabine/Cisplatinum and MVAC being the most frequently used schedules analogous to MIBC [23]. Future trials will need to assess standardized approaches in terms of therapy and the number of cycles.

There was no evidence for significant publication bias according to the ROBINS-I tool. Egger’s graph and funnel plots were positive for two endpoints, as depicted in the Results section (5-year DSS and IPTW adjusted OS). Thus, these results have to be interpreted with care.

## 6. Conclusions

NAC before RNU in UTUC is effective in pathological downstaging and seems to have some oncological benefit in terms of OS and PFS. Exact therapy regimens and patient characteristics to identify those profiting are unclear. Randomized controlled prospective trials are needed to determine whether NAC could become part of a standardized approach. Stratification for parameters like positive locoregional lymph nodes and radiological tumor stage could help to establish selection criteria.

Aside from clinical trials, neoadjuvant immunotherapy does not play a role in the treatment of high-risk UTUC at the moment.

## Figures and Tables

**Figure 1 cancers-14-04841-f001:**
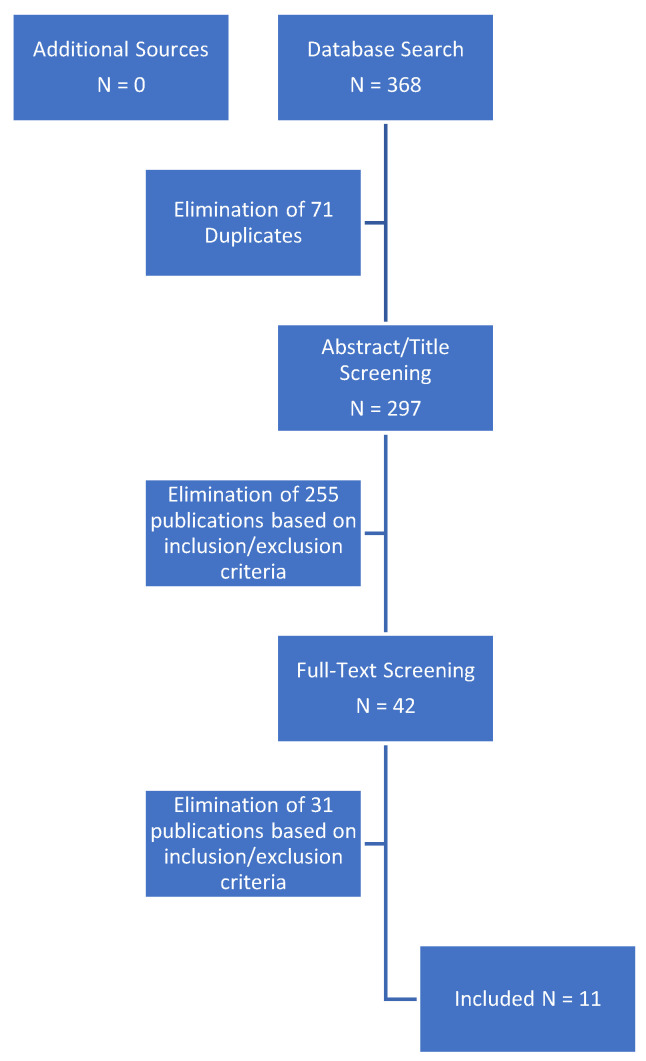
Flowchart of study selection process.

**Figure 2 cancers-14-04841-f002:**
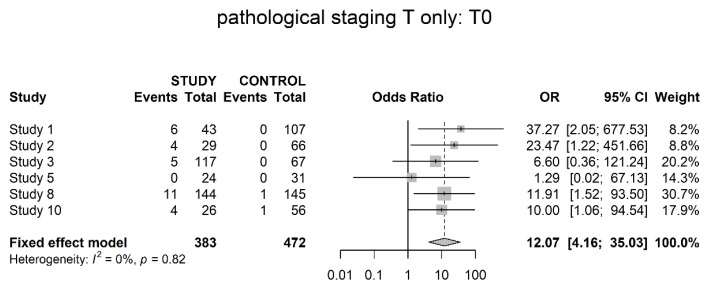
Pathological stage T0 (pathological complete response).

**Figure 3 cancers-14-04841-f003:**
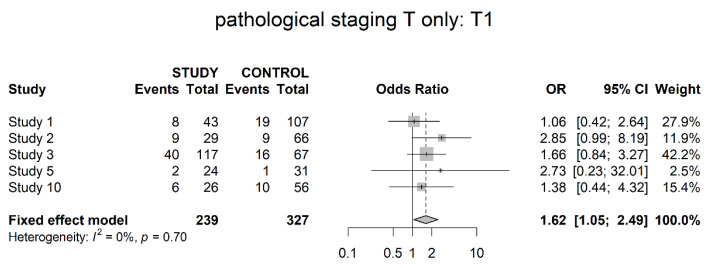
Pathological stage T1 (non-muscle invasive).

**Figure 4 cancers-14-04841-f004:**
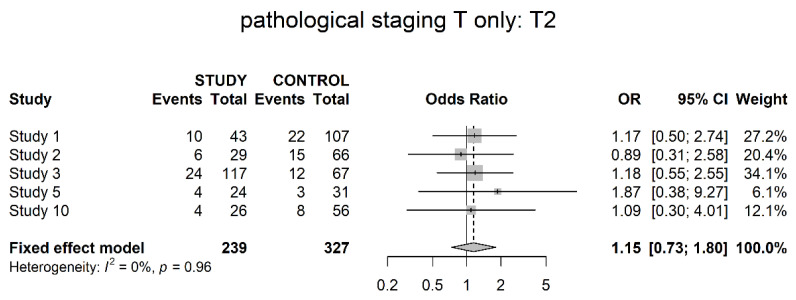
Pathological stage T2 (muscle-invasive).

**Figure 5 cancers-14-04841-f005:**
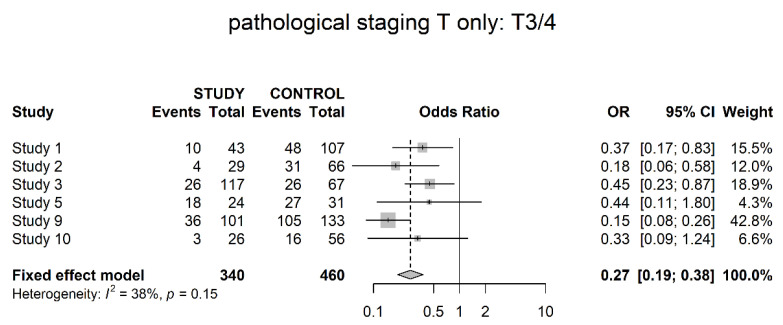
Pathological stage T3/4 (locally advanced).

**Figure 6 cancers-14-04841-f006:**
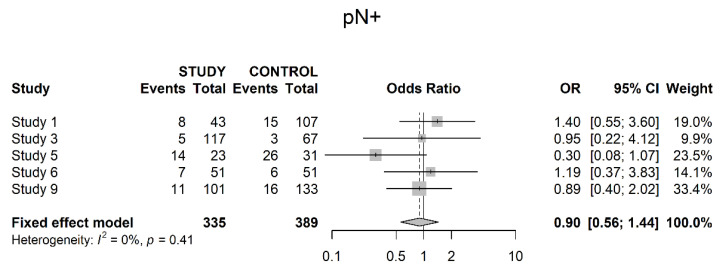
Pathologically positive locoregional lymph nodes.

**Figure 7 cancers-14-04841-f007:**
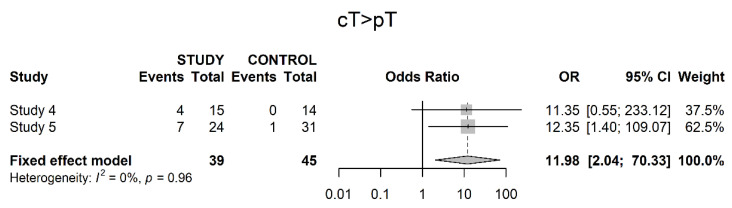
Downstaging from clinical to pathological T stage.

**Figure 8 cancers-14-04841-f008:**
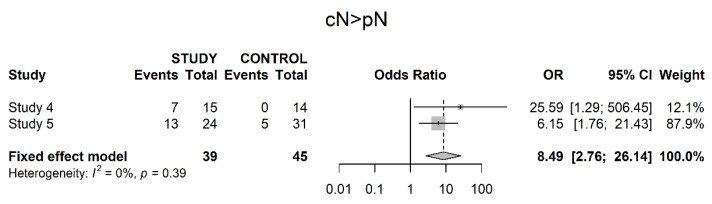
Downstaging from clinical to pathological lymph node stage.

**Figure 9 cancers-14-04841-f009:**
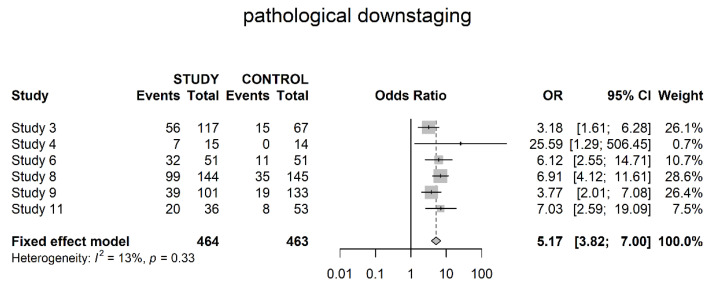
Overall pathological downstaging.

**Figure 10 cancers-14-04841-f010:**
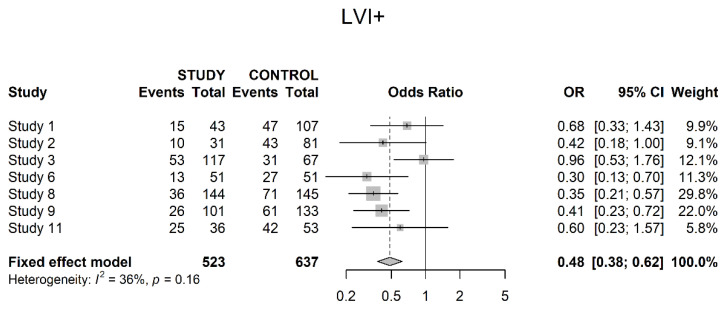
Lymphovascular invasion.

**Figure 11 cancers-14-04841-f011:**
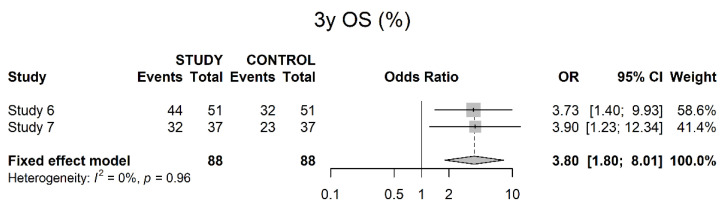
Three-year overall survival.

**Figure 12 cancers-14-04841-f012:**
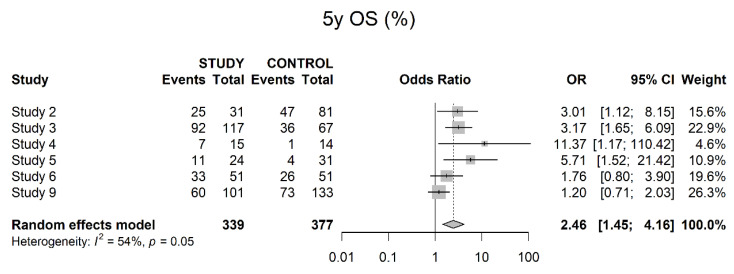
Five-year overall survival.

**Figure 13 cancers-14-04841-f013:**
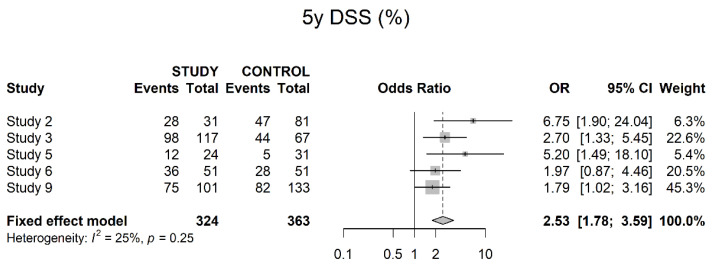
Five-year disease-specific survival.

**Figure 14 cancers-14-04841-f014:**
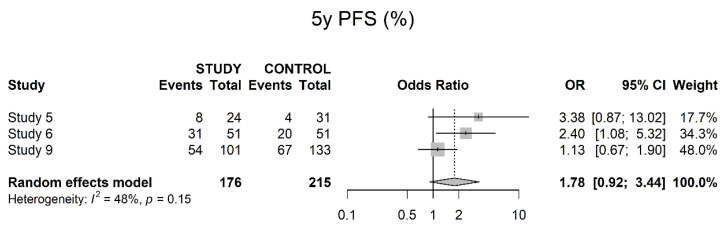
Five-year progression-free survival.

**Figure 15 cancers-14-04841-f015:**
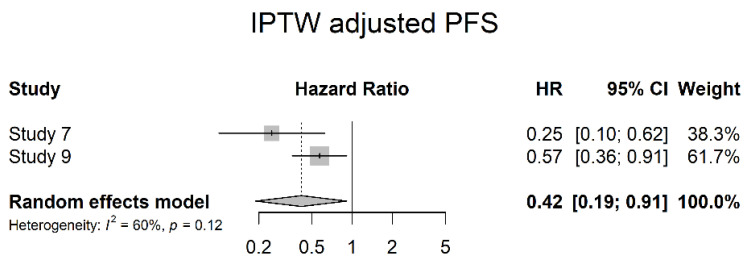
IPTW adjusted progression-free survival.

**Figure 16 cancers-14-04841-f016:**
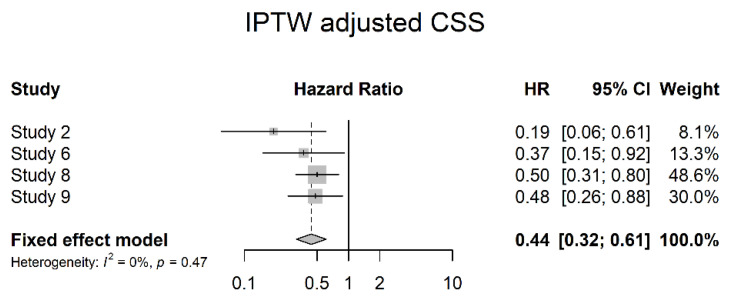
IPTW adjusted cancer specific survival (CSS).

**Figure 17 cancers-14-04841-f017:**
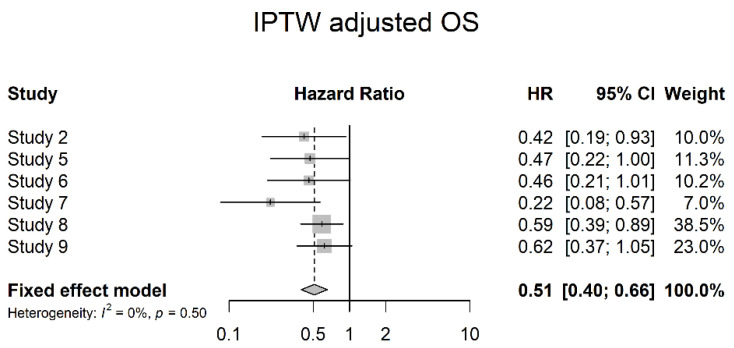
IPTW adjusted overall survival.

**Table 1 cancers-14-04841-t001:** Basic study characteristics.

Study	Authors	Journal	Year	Study Type	Timeline	Pair-Matching	Agent	Cycles	N	Staging Modalities	cN+
1 [55]	Matin et al.	Cancer	2010	retrospective single-center	2004–2008	no	MVAC (19), CGI (9), GTA (6), GC (5), other (4)	2–9 (median 4)	43	CT (Recist Criteria)	no
					1993–2004				107		
2 [56]	Porten et al.	Cancer	2014	retrospective single-center	2004–2008	yes	Cisplatin containing (MVAC, Gem/Cis, Cis/Gem/Ifos; 21), high dose ifosf/doxo/gem (3), kidney sparing (primarily gem/pac/doxo; 7)	2–6 (median 4)	31		no
					1993–2003				81		
3 [57]	Zennamni et al.	BJU Int.	2021	retrospective single-center	2005–2019	yes	Gem/Cis, MVAC	2	117	CT	no
									67		
4 [58]	Kitamura et al.	Jpn J Clin Oncol	2012	retrospective single-center	1995–2010	no	MVAC (14), Gem/Cis (1)	2–3 (median 2)	15	CT (Recist Criteria)	yes (only)
									14		
5 [59]	Kobayashi et al.	Int J Urol	2016	retrospective single-center	1991–2013	no	1991–1995 MEP (3), 1996–2009 MVAC (9), 2010- Gem/Cis (14)	2–4 (median 3)	24	CT (Recist Criteria)	yes (only)
									31		
6 [60]	Hosogoe et al.	Eur Urol Focus	2018	retrospective single-center	1995–2016	yes	Gem/Cis (16), Gem/Carbo (35)	2–4 q3w	51	CT (Recist Criteria)	yes
									51		
7 [61]	Chen et al.	Ann Surg Oncol	2020	retrospective multi-center	2012–2015	yes	Gem/Cis	2–4 q3w	37	CT (Recist Criteria)	no
									37		
8 [62]	Hamaya et al.	BJU Int	2021	retrospective multi-center	2000–2020	no	Gem/Cis, Gem/Carbo, MVAC, docetaxel-based regimen	2–4	144	CT (Recist Criteria)	yes
									145		
9 [63]	Kubota et al.	Onco-target	2017	retrospective multi-center	1995–2017	no	Gem/Cis (21), Gem/Carbo (76), MVAC (4)	2–4 q3w	101	CT (Recist Criteria)	yes
									133		
10 [64]	Rajput et al.	Urology	2011	retrospective single-center	2003–2010	no	variable (MVAC, CGI, MVAC + Bevacizumab, GTA, IAG, CG, GT)	1–7 (median 4)	26	CT or MRI	no
									56		
11 [65]	Shigeta et al.	Urol Oncol	2021	retrospective single-center	1990–2016	no	GC (25), MVAC (11)		36	CT (Recist Criteria) or MRI	yes (only)
									53		

**Table 2 cancers-14-04841-t002:** Descriptive presentation of patient characteristics.

		Study	Control
**cT3/T4**	n_total_	229	246
	n_average_	38	41
	# of studies	6	6
	%_weighted_	84.90%	87.22%
**cT2**	n_total_	74	57
	n_average_	12	10
	# of studies	6	6
	%_weighted_	40.84%	37.59%
**c T1**	n_total_	40	27
	n_average_	10	7
	# of studies	4	4
	%_weighted_	31.13%	32.92%
**cN+**	n_total_	132	134
	n_average_	22	22
	# of studies	6	6
	%_weighted_	65.65%	76.12%
**surgery: open**	n_total_	188	336
	n_average_	38	67
	# of studies	5	5
	%_weighted_	76.76%	89.96%
**surgery: laparoscopic/robotic**	n_total_	67	39
	n_average_	13	8
	# of studies	5	5
	%_weighted_	34.79%	13.49%

Abbreviation: c (clinical staging).

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
