# Peer review of "Neoadjuvant Chemotherapy before Nephroureterectomy in High-Risk Upper Tract Urothelial Cancer: A Systematic Review and Meta-Analysis"

_cancers, 2022, doi:10.3390/cancers14194841_

Round 1
Reviewer 1 Report
This article is a systematic review and meta-analysis, according PRISMA consensus statement, of available date on neoadjuvant treatment for UTUC. This is a very interesting and important work for a rare disease, treated as urothelial tumours of bladder origin, and with few prospective studies directly dedicated to them.
This analysis was carried out using a rigorous methodology. The lack of prospective study and arm control is regrettable but not because of the authors but because of the poverty of studies on this type of tumour. The interpretation of downstaging is to be taken with great caution in these retrospective studies. Only the presence of pT0 on the surgical specimen is a reliable criterion, together with DFS and OS. Limitations are however well described.
One another minor comment : In the introduction, reference 29 should be moderate, only 31 patients have been analysed with WES, which is quiet small to be able to classified them in 4 different clusters, which could explained contradictory results.
Author Response
point 1:
As you mentioned, the lack of prospective data made it difficult to draw tangible conclusions. Aside from reproducible results such as pathological complete response (T0) we tried to collect additional reported measures, such as LVI, reported downstaging et cetera. But of course these measures can not serve as stand-alone surrogates for response. As you mentioned, we tried to discuss this issue properly. However our findings might help to design prospective studies with fitting stratifications to identify appropriate populations for neoadjuvant therapy.
point 2:
This is a very good explanation for the contradictory results within the cited studies. I added a remark in the text explaining the low number of samples analyzed. I did not cut out the reference completely, since I believe it is interesting to see the different data (of different quality). Especially because the related question (response to immunotherapy in UTUC) is not answered sufficiently yet. It might be a good idea to repeat the analysis in a larger sample in a future trial.
Reviewer 2 Report
This report is a well-written meta-analysis depicting promising effects of neoadjuvant chemotherapy (NAC) for upper tract urothelial cancer on downstaging and survival outcomes. Various concerning issues need to be addressed prior to publication.
Comments:
In the Pathological Downstaging section, authors demonstrate the pathological T1 and T2 proportion in both control and study groups. It would be imperative to compare T0 (complete response) rate in both groups. It would also be crucial to demonstrate that NAC may reduce advanced disease. However, the purpose of the evaluation of pathological T1 and T2 is obscure.
In the Discussion section, the authors should elaborate more on the findings that are added by this meta-analysis to the latest comprehensive meta-analysis by Leow et al. besides the inclusion of four new studies.
Author Response
point 1:
Due to the lack of prospective data, heterogenous study designs and especially the intrinsic diagnostic and therapeutic characteristics of UTUC, it was difficult to find tangible data on the issue. Therefore we tried to analyze multiple additional parameters reported throughout the studies, in order to obtain further insights.
As you mentioned pathological complete response (T0) is a clear measure of complete response to neoadjuvant therapy. Furthermore T0 state is a surrogate for improved OS in bladder cancer. On the other hand locally advanced tumor stages are associated with worse outcomes. Analysis thereof are important to get an impression of reduction of advanced stages by neoadjuvant therapy, but also progress under chemotherapy might be assessed.
Regarding the raised question, the stages in-between T1 (non-muscle invasive) and T2 (muscle-invasive, but not locally advanced) are difficult to use as surrogate markers for response. However they might give an impression on partial response. T1 disease (or lower) has already been used as a surrogate marker for partial response; amongst others in the systematic review of Leow et al. We descriptively depicted clinical staging data before therapy (Table 2), which seemed balanced. Based on that our finding of significantly higher numbers of T1 tumors within the therapy group might indicate higher response. We were not able to show any differences within the T2 group, however we included the data for a comprehensive depiction of the pathological stages throughout both study groups.
We are fully aware of the low quality of data and the various confounders. We tried to discuss the various limitations extensively and we strongly recommend to interpret these findings with care. Concerning T stage and partial response the major problem is that there is no reporting of individual patient data with change of clinical to pathological T stage. The only surrogate finding for this measure, which was presented in the studies was "pathological downstaging", which we analyzed separately. Very few studies also explicitly presented downstaging of N or T stage. Based on these findings we want to suggest more thorough study designs in the future, adressing the issue of partial response. This might even be feasible for retrospective study designs, in which exact presentation of individual-based staging data before and after treatment in both groups might compensate for some selection bias.
In order to clarify our intentions on reporting T1 and T2 data I added an explanation to the discussion.
point 2:
I added a paragraph elaborating on the differences in the two analysis, additional parameters we looked at as well as new findings of our meta-analysis.